# Effectiveness of Preventive Measures in Keeping Low Prevalence of SARS-CoV-2 Infection in Health Care Workers in a Referral Children's Hospital in Southern Italy

**Désirée Caselli** [1,†] , **Daniela Loconsole** [2,†] , **Rita Dario** [3] , **Maria Chironna** [2,‡] and **Maurizio Aricò** [3,*,‡]

1   Pediatric Infectious Diseases, Giovanni XXIII Children Hospital, Azienda Ospedaliero Universitaria Consorziale Policlinico, 70126 Bari, Italy; desiree.caselli@policlinico.ba.it
2   Department of Biomedical Sciences and Human Oncology-Hygiene Section, University of Bari, 70120 Bari, Italy; daniela.loconsole@uniba.it (D.L.); maria.chironna@uniba.it (M.C.)
3   COVID Emergency Task Force, Giovanni XXIII Children Hospital, Azienda Ospedaliero Universitaria Consorziale Policlinico, 70126 Bari, Italy; rita.dario@policlinico.ba.it
*   Correspondence: maurizio.arico@policlinico.ba.it
†   D.C. and D.L. equally contributed as first authors.
‡   M.C. and M.A. equally contributed as senior authors.

**Abstract:** The coronavirus disease 2019 (COVID-19) pandemic now represents a major threat to public health. Health care workers (HCW) are exposed to biological risk. Little is currently known about the risk of HCW operating in pediatric wards for SARS-CoV-2 infection. The aim is to assess the prevalence of SARS-CoV-2 infection in HCW in a third-level children's hospital in Southern Italy. An observational cohort study of all asymptomatic HCW (physician, technicians, nurses, and logistic and support operators) was conducted. HCW were screened, on a voluntary basis, for SARS-CoV-2 by RT-PCR on nasopharyngeal swab performed during the first wave of COVID-19. The study was then repeated, with the same modalities, at a 7-month interval, during the "second wave" of the COVID-19 pandemic. At the initial screening between 7 and 24 April 2020, 525 HCW were tested. None of them tested positive. At the repeated screening, conducted between 9 and 20 November 2020, 627 HCW were tested, including 61 additional ones resulting from COVID-emergency recruitment. At this second screening, eight subjects (1.3%) tested positive, thus being diagnosed as asymptomatic carriers of SARS-CoV-2. They were one physician, five nurses, and two HCW from the logistic/support services. They were employed in eight different wards/services. In all cases, the epidemiological investigation showed convincing evidence that the infection was acquired through social contacts. The study revealed a very low circulation of SARS-CoV-2 infection in HCW tested with RT-PCR. All the infections documented in the second wave of epidemic of SARS-CoV-2 were acquired outside of the workplace, confirming that in a pediatric hospital setting, HCW education, correct use of personal protective equipment, and separation of the COVID-patient pathway and staff flow may minimize the risk derived from occupational exposure.

**Keywords:** health care workers; SARS-CoV-2; COVID-19; screening





## 1. Introduction

Italy has been one of the most affected European countries during the first wave of the coronavirus 2019 (COVID-19) pandemic [1], which in our country spanned between January 2020 and beginning of June 2020. Italy was the first to institute a national lockdown to contain the spread of the virus, effective 9 March 2020. School closure was decided some days before the beginning of the lockdown and started from 5 March 2020. A second wave of the pandemic progressively arose starting from mid-August 2020 and is still ongoing at the time of writing. As of 3 February 2021, the pandemic has resulted in 2,579,763 cases and 88,533 deaths in Italy [2].

Apulia is a large region in Southern Italy, with about four million inhabitants, including 652,754 children. The proportion of inhabitants diagnosed with COVID-19 has been lower in Apulia than in regions of northern Italy during the so-called "first wave". The first case was identified on 26 February 2020, in a 44-year-old man in Taranto province after arrival from a known COVID-19 hotspot (Lodi) in Northern Italy. As of 1 June 2020, 4498 cases of confirmed SARS-CoV-2 infections were reported in Apulia [3]. The picture became completely different during the course of the year 2020, with the so-called "second wave": as of 4 February 2021, a total number of 1,346,440 diagnostic tests have identified 126,184 positive cases in Apulia region [4].

The main route of SARS-CoV-2 transmission is airborne transmission, in which aerosol transmission encompasses the transfer of pathogens via droplet nuclei. Accordingly, person-to-person contact may enhance transmission, and asymptomatic and pre-symptomatic subjects play an essential role in spreading the infection [5].

Healthcare workers (HCW) are a vulnerable cohort for infection, due to frequent and close contact to COVID-19 patients [6]. In turn, HCW could represent a source of infection for hospital susceptible patients and co-workers [7]. The reported transmission rates in healthcare settings are variable [8–11]. Since the beginning of the pandemic, national and local guidelines have been implemented to reduce the transmission of SARS-CoV-2 among HCW, including adherence to strict hygiene standards, using suitable personal protective equipment (PPE) and applying social distancing, including outside the hospital [12–15]. A proactive SARS-CoV-2 screening program for the HCW of "Giovanni XXIII" Pediatric Hospital of Bari was implemented in order to quantify the circulation of SARS-CoV-2 among asymptomatic HCW and to create a safe hospital environment for staff and patients. Here, we report the results of a two-round screening activity.

## 2. Materials and Methods

The "Giovanni XXIII" Pediatric Hospital of Bari is a teaching institution, serving as a referral for the entire Apulia region, Southern Italy. It encompasses 12 wards and 156 beds for inpatients.

In order to protect all exposed HCW from the risk of contracting SARS-CoV-2 infection following exposure to patient and his/her accompanying relative/s, but also from any potential asymptomatic carrier, the hospital defined its bylaws in terms of personal protective equipment (PPE) use. The minimal composition of the set of PPE for the management of suspected case of COVID-19 was respiratory protection with FFP2 (with extended use over a single daily shift), long-sleeved water-resistant gown, gloves, goggles, or face-shield. All HCW were trained on dressing and undressing the PPE properly. The use of gloves and surgical mask was allowed for activities not in the presence of a patient at a short distance. The use of the most protective set of PPE was restricted to the COVID area and to procedures generating aerosol [16].

The hospital management implemented a two-round voluntary SARS-CoV-2 testing program for the HCW. Indeed, with the aim of evaluating the entire hospital staff, the screening was open not only to all HCW (i.e., those workers who directly cared patients) but also to technical and administrative staff, and even to "support facilities" personnel working within the hospital. Thus, for the purpose of this study, the term HCW includes this enlarged population.

All HCW were informed about the protocol and its purposes. All subjects reporting symptoms potentially related to COVID-19, or in close contact with a COVID-19 patient, were excluded from this screening since they were asked to immediately notify the hospital authority and thus followed another specific procedure.

Nasopharyngeal swabs were collected from all employees over a 2-week testing window: between 7 and 24 April 2020, during the initial wave of the pandemic (screening 1), and between 9 and 20 November 2020, during the so-called second wave (screening 2). During each round of screening, the nasopharyngeal swab was performed by a single operator, over five working days (Monday to Friday), in two consecutive weeks.

The following wards were defined as "COVID-19 Area": infectious diseases, emergency room, and intensive care unit.

Staff was permitted to return to work while awaiting test results. All HCW positive for SARS-CoV-2 infection were to be excluded from work and subjected to 14-day administrative leave (during screening 1) or 10-day administrative leave (during screening 2) according to current bylaws. The change in duration of the isolation measure was due to new indications from the Ministry of Health in October 2020 [17]. An epidemiological investigation in search of contacts was initiated for all cases. After the isolation period, the subject was re-tested.

All samples were analyzed at the Laboratory of Molecular Epidemiology and Public Health of the Hygiene Unit (A.O.U.C. Policlinico Bari), which is the coordinating center of the Regional Laboratory Network for SARS-CoV-2 diagnosis. The presence of E gene, RdRP gene, and N gene of SARS-CoV-2 virus was identified by a commercial real-time PCR assay (Allplex2019-nCoV Assay; Seegene, Seoul, Korea). The results were available the day after the screening.

### 3. Results

In the intention to cover with our screening the entire population of operators acting around children within the hospital, we included as "HCW" not only those who directly cared patients, but also operators from the technical, administrative, and logistic support areas.

In the first screening, this population accounted for 616 subjects. Of them, 91 were on either smart working or leave (personal reason or "fragile subject"). The remaining 525 (85%) were actually on service at the time of the screening and were included, with no refusal. Their mean age was 50.3 years. Of them, 135 were physicians, biologists, or psychologists; 245 were nurses or technicians; 33 were auxiliary nurses; 64 were in support services; and 48 were in the logistics. None of them tested positive at nasopharyngeal swab.

At the repeated screening, the study population accounted for 677 subjects. Of them, 50 were on either smart working or leave (personal reason or "fragile subject"). The remaining 627 were actually on service at the time of the screening and were included, with no refusal. Their mean age was 45.6 years. Of them, 143 were physicians, biologists, or psychologists; 317 were nurses or technicians; 41 were auxiliary nurses; 74 were in support services; and 52 were in the logistics. The number includes 61 additional HCW (especially nurses) resulting from COVID-emergency recruitment. At this round, eight operators (1.3%) tested positive at RT-PCR on nasopharyngeal swab, thus being diagnosed as asymptomatic carrier of SARS-CoV-2. Of the eight positive HCW, one was a physician and five were nurses. Remarkably, they were from different wards and only one nurse was from the COVID area. Main results are summarized in Table 1.

**Table 1.** Number and distribution of health care workers (HCW) screened.

| | Screening 1 (n = 525/616; 85%) | | Screening 2 (n = 627/677; 93%) | |
|---|---|---|---|---|
| | **Number Negative** | **Positive** | **Number Negative** | **Positive** |
| Total HCW screened | 525 | 0 | 619 | 8 (1.3%) |
| Medical | | | | |
| COVID Area * | 25 | | 38 | |
| Non COVID Area | 109 | | 116 | 1 |
| Nursing | | | | |
| COVID Area * | 63 | | 82 | 1 |
| Non COVID Area | 157 | | 192 | 4 |
| Administrative | 88 | | 98 | |
| Support service | 83 | | 93 | 2 |

* COVID area includes infectious diseases, emergency room, and intensive care.

## 4. Discussion

We have progressively learned that the SARS-CoV-2 and the COVID-19 pandemic, for reasons that have not yet been fully clarified, largely spared children and adolescents worldwide, compared to the aggressiveness and lethality rate observed in adults [18–21]. Since children and adolescents had milder manifestations of the disease, their need of hospital admission was minimal. As a consequence, during the "first wave" of the pandemic, i.e., during the first six months of 2020, our children's hospital was only marginally involved in the tide of patients with COVID-19, and a only very small number of beds in the Infectious diseases ward, never more than 2–3, were usually occupied by COVID-19 in-patients. Nevertheless, the fear that HCW would be infected was very high. Thus, we chose to assess the infection status for SARS-CoV-2 of all our operators, whether they were directly caring for children or only indirectly supporting in health care or even logistics in the hospital. None of our HCW refused to undergo a voluntary screening, a finding that is in keeping with data reported by Oster et al., although in contrast with other studies, which reported percentages of adhesion to the test lower than they expected [12,22]. It is noteworthy that at that time healthy subjects in our country had no access to performing nasopharyngeal swab for assessment of SARS-CoV-2 infection status. Thus, the screening offered was considered an attractive opportunity to be reassured by a negative test.

We decided to use RT-PCR on nasopharyngeal swab and not serology for the screening of our HCW. The limit of PCR-based screening is that the results should be considered valid only for the day of the test, and this may cause a false sense of confidence. On the other hand, we considered a screening based on antibodies anti SARS-CoV-2 to be inappropriate for our purposes, since it could not give information about the current infectivity of HCW, which could represent a source of infection for patients and colleagues. The seroprevalence could only give information about the application of hygiene measures and the awareness of COVID-19 transmission. It is interesting to note that in a recent report from a north-American multi-state hospital network, of 156 HCW who tested seropositive at SARS-CoV-2 in spring 2020, a substantial proportion might have had negative serologic test results at the 60-day re-test [23].

Based on the reassuring negative result, we decided not to repeat the screening during the following few months. During the summer, the lock-down policy was dismissed and the life-style of the population returned to almost normal. In particular, since Apulia has a clear vocation to leisure and tourism, tens of thousands people visited our region, either to meet relatives in their place of origin or to spend their summer vacation. This was associated with a sharp increase of the pandemic, starting from the last week of August, which then turned into a second tide by mid-October 2020. Therefore, we observed a

rapid change in the epidemiology of the disease in our area. In particular, starting from September 2020, we faced a completely different number of children with COVID-19 coming to our children's hospital, with the number of beds usually occupied by COVID-19 children ranging between 6 and 10. In the meanwhile, widespread SARS-CoV-2 nosocomial infections had been reported by hospitals worldwide [24]. Holmes et al. reported a 12.5% viral transmission rate amongst orthopedic patients despite the segregation measures taken, possibly due to asymptomatic HCW or inpatients awaiting swab results [25]. Thus, circulation of SARS-CoV-2 among asymptomatic HCW became a continuous worry for HCW themselves, contributing to their not only physical but also psychological stress. In this setting, we decided to offer our employees a second screening, which was very well received by HCW. We registered a very low proportion of asymptomatic carriers (only 1.3%). This value is comparable with that reported by Oster et al. [12] but much lower to what reported in other settings. Among them, in a large study of 12,541 HCW from England and Vietnam who had anti-spike IgG measured, 100 asymptomatic and seronegative HCW had a positive PCR test [26]; in a serological study of over 6000 HCW in Spain, when adding PCR test, 713 (11.8%) employees showed evidence of SARS-CoV-2 infection [27]. Altogether, these data support the concept that circulation of SARS-CoV-2 among asymptomatic HCW in the hospital setting may vary widely. In an interesting study of perceived versus proven SARS-CoV-2-specific immune responses in HCW, study participants estimated their personal likelihood of having had a SARS-CoV-2 infection at a mean of 21%, while their specific seroprevalence was about 1–2% [28]. Available data on circulation of SARS-CoV-2 among asymptomatic HCW in children's hospitals are limited; in a seroprevalence study of physicians from a children's hospital in Argentina, Insúa et al. observed that only 0.9% were seropositive [29].

It is noteworthy that only six of the 433 (1.38%) medical and nursing HCW tested positive, with only one nurse from the COVID wards. Furthermore, all of the eight positive HCW were from different wards/services. Thus, no cluster effect was observed. We consider this result very rewarding for the hospital organization, inasmuch as, despite being a large institution, which never stopped providing emergency services, as well as programmed care of children with severe/chronic conditions, we have been able to protect our HCW from professional exposure to SARS-CoV-2. The low percentage of positive HCW is likely due to the following factors. First, the huge effort made by the hospital governance to continuously provide the necessary PPE to any employee, according to the risk-directed equipment policy. Moreover, once the PPE had been provided, the policy to wear them was strongly enforced by the hospital management. All HCW were adequately educated about the correct donning and doffing procedures, before and after any contact with patients. This was probably the main reason for the very limited proportion of positive subjects observed even at the second screening.

To prevent children from SARS-CoV-2 exposure, we designed and implemented two specific pathways, aimed at separating the inpatients since admission, according to their SARS-CoV-2 infection status. At the front desk, all patients are asked to report their social/familial risk of exposure to a person diagnosed with COVID-19. Furthermore, the presence of clinical manifestations was typical of COVID-19 in children (i.e., in particular persistent fever >38 °C with respiratory syndrome). If so, they were addressed to the COVID-pathway, consisting of a dedicated Fast Track for initial evaluation, aside from and independent of the emergency room. From 20 April 2020, all patients deserving admission to the ward (with the only exception of true emergencies) went through a "Grey Area" in which a nasopharyngeal swab was performed. Upon evidence of negativity at this test, the patient was released to the competent ward based on the reason of admission. Otherwise, all patients already known to be SARS-CoV-2 positive by RT-PCR test, or resulting positive at the nasopharyngeal swab screening, were admitted to the "Red Area". Both the Grey and the Red areas are part of the Infectious Diseases department of the children's hospital. This setting allowed us to completely separate patients negative for SARS-CoV-2, and

thus to define a COVID-free hospital, from those who were waiting for testing results or tested positive.

The detailed evaluation of the eight asymptomatic, SARS-CoV-2-positive employees clearly shows that there was no cluster effect. They worked in different wards and services, as either physicians, nurses, or in logistics. Tracing provided convincing evidence that the source of their infection was a social contact and thus excluded professional circulation of SARS-CoV-2.

We conclude that the strategy adopted and the organization of our children's hospital proved to be safe and effective in protecting the HCW from SARS-CoV-2 professional exposure. Furthermore, it proved also protective for the children and their family, inasmuch as all children requesting emergency or programmed care were accepted and treated, although only by using strict measures to identify SARS-CoV-2 carriers among patients and their caring parents. Such measures have been continuously enforced so far; we hope the vaccination machinery, which has just been implemented, will be able to progressively build a protective umbrella for our population, including young patients and HCW.

**Author Contributions:** Conceptualization, D.C., D.L., M.C. and M.A.; Data curation, D.L. and R.D.; Investigation, D.C., M.C. and M.A.; Methodology, D.L. and R.D.; Supervision, D.C., M.C. and M.A.; Writing—original draft, M.A.; Writing—review & editing, D.C., R.D. and M.C. All authors have read and agreed to the published version of the manuscript.

**Funding:** This research received no external funding.

**Institutional Review Board Statement:** The study was conducted according to the guidelines of the Declaration of Helsinki; Institutional review Board review and approval were waived for this study, since it reported anonymized data resulting from an activity decided directly by the Hospital General Manager.

**Informed Consent Statement:** Informed consent was obtained from all subjects involved in the study.

**Data Availability Statement:** The anonimyzed data presented in this study are available on request from the corresponding author.

**Conflicts of Interest:** The authors declare no conflict of interest.

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
