# Peer review of "Effectiveness of Preventive Measures in Keeping Low Prevalence of SARS-CoV-2 Infection in Health Care Workers in a Referral Children’s Hospital in Southern Italy"

_pediatrrep, doi:10.3390/pediatric13010017_

Round 1

Reviewer 1 Report

I have read with interest the paper titled “Effectiveness of preventive measures in keeping low prevalence of SARS-CoV-2 infection in Health Care Workers in a referral children hospital in Southern Italy” submitted for publication to Pediatric Reports. There, authors present the results of a swab screening in a teaching hospital in Italy.

On the whole, the paper needs some improvements to be accepted for publication. Here, please find my comments and suggestions. English should be improved too.

Introduction

- Authors correctly included some surveillance data, but these should be homogeneous and referred to the same time-point. Indeed, data are reported as of June 16 for Italy and as of Jan 2021 for Apulia, making difficult to the reader to understand the context epidemiological.

- When talking about “first wave”, please include the temporal window this refers. Same for the second wave in Methods.

- Technically speaking, the route of SARS-CoV-2 and disease transmission is not person-to-person contact, but an “Airborne Transmission”, in which aerosol transmission encompasses the transfer of pathogens via droplet nuclei. So that, person-to-person contact enhance the transmission, but this is NOT a route.

- I have doubts regarding the sentence “A vulnerable cohort for infection due to frequent and close contact to COVID-19 patients are healthcare workers (HCWs)”. Although apparently correct, HCWs are not “vulnerable for infection” (the whole general population is), they are “just” at increased exposure to the virus. Briefly, the risk (RR) is the same for everyone, but not the exposure.

- Please, remove the Web links from the text. These are references.

Methods.

- Reasons for exclusion of “All subjects re-porting symptoms potentially related to COVID-19, or close contact with a COVID-19 patient, were excluded”.

- Documento of Minis-try of Health 0032850-12/10/2020-DGPRE-DGPRE-P is a refs.

- “which is the coordinator of the Regional Laboratory Network for SARS-CoV-2 diagnosis.” I’d rather say coordinating center; coordinator refers to the person who coordinates.

- The complete definition of HCW in the paper is not correct, since authors also include technical and administrative staff, i.e., non-HCWs. The definition of HCWs includes in general only those workers who do care patients; while all the others are classified as “non-HCWs”. Both groups constitute a hospital staff.

- Definition of COVID-19 Area should be included in the Methods section

Discussion

- The whole first two sentences should be rephrased for clarification. In particular, what does “with never more than 2-3 patients with COVID-19 simultaneously treated as in-patients.” mean?

- Authors enthusiastically wrote about an important compliance of the hospital staff towards the screening, but the whole hospital worker population nor the response rate were reported in methods and results.

- “This is in accordance with data reported by Oster et al., but in contrast with data of Jameson et al., that reported a percentage of adhesion to the test lower than they expected (10,14).” Something more about these comparisons should be added.

Briefly, the whole article is not innovative enough. The analysis was too simple, and the differences between the time lags were obvious. It is recommended that the authors do more comparative analysis on other countries, to see whether the differences between children and non-children hospital exist, both in Italy and elsewhere (on the basis of available evidence).

Author Response

Reviewer 1 - Comments and suggestions.

Introduction

  • Authors correctly included some surveillance data, but these should be homogeneous and referred to the same time-point. Indeed, data are reported as of June 16 for Italy and as of Jan 2021 for Apulia, making difficult to the reader to understand the context epidemiological.
    • National epidemiological data were updated at the time of writing.
  • When talking about “first wave”, please include the temporal window this refers. Same for the second wave in Methods.
    • The dates for the first and the second waves have been indicated already in the introduction.
  • Technically speaking, the route of SARS-CoV-2 and disease transmission is not person-to-person contact, but an “Airborne Transmission”, in which aerosol transmission encompasses the transfer of pathogens via droplet nuclei. So that, person-to-person contact enhance the transmission, but this is NOT a route.
    • We acknowledge that the statement was not fully correct. It was modified according to your remarks.
  • I have doubts regarding the sentence “A vulnerable cohort for infection due to frequent and close contact to COVID-19 patients are healthcare workers (HCWs)”. Although apparently correct, HCWs are not “vulnerable for infection” (the whole general population is), they are “just” at increased exposure to the virus. Briefly, the risk (RR) is the same for everyone, but not the exposure.
    • We acknowledge that the statement was not fully correct. It was modified according to your remarks.
  • Please, remove the Web links from the text. These are references.
    • This was done.

 Methods.

  • Reasons for exclusion of “All subjects re-porting symptoms potentially related to COVID-19, or close contact with a COVID-19 patient, were excluded”.
    • The screening was aimed at identifying SARS-CoV-2 among apparently those HCW who appeared as “healthy, non-risk subjects”. HCW with symptoms (none in the first wave) or under tracking because of a “close contact” (again very little numbers during the first and also the second wave), were picked-up by the COVID Control Room and followed differently.
  • Documento of Ministry of Health 0032850-12/10/2020-DGPRE-DGPRE-P is a refs.
    • This was changed.
      • “which is the coordinator of the Regional Laboratory Network for SARS-CoV-2 diagnosis.” I’d rather say coordinating center; coordinator refers to the person who coordinates.
    • We acknowledge that the statement was not correct. It was modified according to your remarks.
      • The complete definition of HCW in the paper is not correct, since authors also include technical and administrative staff, i.e., non-HCWs. The definition of HCWs includes in general only those workers who do care patients; while all the others are classified as “non-HCWs”. Both groups constitute a hospital staff.
    • We acknowledge that the statement was not correct. The definition was modified, to be more detailed and clarify.
      • Definition of COVID-19 Area should be included in the Methods section
    • The definition of “COVID-19 Area” was clarified in the Methods section as suggested.

Discussion

  • The whole first two sentences should be rephrased for clarification. In particular, what does “with never more than 2-3 patients with COVID-19 simultaneously treated as in-patients.” mean?
  • We acknowledge that the statement was not fully correct and potentially misleading. It was expanded and clarified as follows: “Since children and adolescents had milder manifestations of the disease, their need of hospital admission was minimal.
  • As a consequence, during the “first wave” of the pandemic, i.e. during the first six months of the year 2020, our children hospital has been only marginally involved in the tide of patients with COVID-19, and a only very small number of beds in the Infectious diseases ward, never more than 2-3, were usually occupied by COVID-19 in-patients. ”
    • Authors enthusiastically wrote about an important compliance of the hospital staff towards the screening, but the whole hospital worker population nor the response rate were reported in methods and results.
  • We acknowledge that the statement could be improved. We have given in the methods the details of the study population, explaining the reasons for selecting this broad version of HCWs. In the first paragraph of the results, the number of the operators who were not tested was also reported, with the reasons. None was a refusal. Furthermore, the denominator was also mentioned in the table. The “enthusiasm” was mitigated in the discussion.
    • “This is in accordance with data reported by Oster et al., but in contrast with data of Jameson et al., that reported a percentage of adhesion to the test lower than they expected (10,14).” Something more about these comparisons should be added.
  • We added a statement addressing the growing observation of nosocomial infections.
  • Briefly, the whole article is not innovative enough. The analysis was too simple, and the differences between the time lags were obvious. It is recommended that the authors do more comparative analysis on other countries, to see whether the differences between children and non-children hospital exist, both in Italy and elsewhere (on the basis of available evidence).
    • We acknowledge that the manuscript could be enriched in this perspective. We introduced in the discussion a comparison with other experiences in hospital HCW screening and with a few available data on children hospital, too.

Reviewer 2 Report

The study is well described, the paper well written. The authors should, however, present the numbers shown on screening results in Table 1 with additional information on the total number of employees in their institution, ideally for the different groups they mention (medical [doctors, nurses, auxiliary], administrative [support & logistics]).

This information could be added to the first paragraph of the Materials and Methods section.

The authors in the Discussion that they observed "full compliance" of the personnel and that "all the eligible HCW [health care workers]" accepted the screening procedure. This could be expressed numerically in the first paragraph of the Results section and/or in Table 1.

In addition - and in order to fully appreciate the measures taken by their institution to protect its HCW staff - the authors should give a somewhat more detailed description in the Methods Section of the personal protective equipment and the way it was used in daily practice (e.g., which kind of masks were available, were they re-used? etc.).

Author Response

  • The study is well described, the paper well written. The authors should, however, present the numbers shown on screening results in Table 1 with additional information on the total number of employees in their institution, ideally for the different groups they mention (medical [doctors, nurses, auxiliary], administrative [support & logistics]). This information could be added to the first paragraph of the Materials and Methods section.
    • We thank the reviewer for this suggestion, fully in keeping with the comment made by reviewer 1. We added these data in the first paragraph of the Materials and Methods section
  • The authors in the Discussion that they observed "full compliance" of the personnel and that "all the eligible HCW [health care workers]" accepted the screening procedure. This could be expressed numerically in the first paragraph of the Results section and/or in Table 1.
    • We agree with the reviewer. These data were introduced in the first paragraph of the results, the number of the operators who were not tested was also reported, with the reasons. None was a refusal. Furthermore, the denominator was also mentioned in the table.
  • In addition - and in order to fully appreciate the measures taken by their institution to protect its HCW staff - the authors should give a somewhat more detailed description in the Methods Section of the personal protective equipment and the way it was used in daily practice (e.g., which kind of masks were available, were they re-used? etc.).
    • This was introduced in the Materials and Methods section.

Round 2

Reviewer 1 Report

Authors have addressed my comments.